# Anatomical Tool as Additional Approach for Identifying Pharmaceutically Important *Ephedra* Species (Ephedraceae) at Gender Identity Level in Egypt

**DOI:** 10.3390/biology13110947

**Published:** 2024-11-18

**Authors:** Maha H. Khalaf, Wafaa M. Amer, Najla A. Al Shaye, Mahmoud O. Hassan, Nasr H. Gomaa

**Affiliations:** 1Department of Botany and Microbiology, Faculty of Science, Beni-Suef University, Beni-Suef 62521, Egypt; maha.hamdy@science.bsu.edu.eg (M.H.K.); dr_mody1983_science@yahoo.co.uk (M.O.H.); nasr.gomaa@science.bsu.edu.eg (N.H.G.); 2Department of Botany and Microbiology, Faculty of Science, Cairo University, Giza 12613, Egypt; wamer@sci.cu.edu.eg; 3Department of Biology, College of Science, Princess Nourah bint Abdulrahman University, P.O. Box 84428, Riyadh 11671, Saudi Arabia

**Keywords:** *Ephedra* fragments, light microscope, stem anatomy, epidermal system, gender identification, Ephedraceae

## Abstract

The genus *Ephedra* holds significant importance in medicinal and environmental fields. Based on morphological characteristics, taxonomists often face challenges in identifying *Ephedra* specimens lacking reproductive cones, which are key features for identification. This study explored stem anatomy as a potential method for identifying *Ephedra* species and distinguishing between male and female plants. To this end, we analyzed the stem anatomy of five *Ephedra* species found in Egypt. Our findings revealed significant differences in these features, including tanniniferous idioblasts, the stem outline, the number of tracheids, and the number of protoxylem arches, allowing for the reliable identification of each species and genders within each species. This new approach provides a valuable tool for identifying *Ephedra* specimens lacking cones, which is important for researchers, pharmacists, and environmentalists who work on these plants. Our findings suggest that a multidisciplinary approach, combining anatomy with morphological characteristics, can effectively identify *Ephedra* specimens with reduced morphological features.

## 1. Introduction

The genus *Ephedra* Tourn. ex L. (Ephedraceae), is a monophyletic group within the order Gnetales [1] and often forms distinct subgroups in the neighboring geographic regions [2,3,4]. Currently, the genus *Ephedra* comprises around 73 accepted species [5] that thrive in open, arid, or semi-arid habitats such as deserts and rocky slopes in Asia, Europe, N. Africa, and western N. and S. America [2,6,7]. *Ephedra* species have significantly reduced vegetative parts [1], notably the reduced, scaly leaves with minimal chlorophyll [6,8,9,10,11,12,13,14,15].

*Ephedra* includes small, drought-resistant, woody sub-shrubs, or herbs, typically not exceeding 4 m in height. The stems are often densely branched, with green branchlets arranged opposite or in whorls at the nodes. The shoots are green assimilating, terete, and articulate, with furrowed internodes. The epidermal cells are highly cuticularized, except in the areas where stomata are located. The stomata are arranged in rows and furrows. The leaves are scale-like, oppositely attached at their base to form a sheath, and deciduous. The *Ephedra* species are monoecious, where the male and female reproductive organs are arranged in cones, with each cone subtended by 2–4 or more pairs of opposite, scale-like bracts [5,16,17,18]. With an interesting anatomical structure, *Ephedra* fills an intermediate position between *Welwitschia* and *Gnetum* in the order Gnetaceae and in the lower rank among Gymnosperms [19].

The epidermal characteristics of gymnosperms are very significantly important for differentiating the gymnosperm taxa [20,21,22]. The cuticular features were used as the sole parameter for identifying the fossil gymnosperms [14]. Cuticular analysis has been utilized as a valuable method for identifying, classifying, and correlating taxa in taxonomic studies focused on stomatal morphology, including stomatal type, size, and index [23,24,25,26,27]. Additionally, the epidermal and stomatal structures of *Ephedra foliata* Boiss. ex C.A. Mey. has been extensively investigated [12,28,29].

Earlier studies carried out on the stem anatomy of Gnetales at the species level aimed to gather new insights into their systematic and phylogenetic significance [30,31,32]. These studies provided detailed anatomical inspections of *Ephedra*, *Gnetum*, and *Welwitschia*, while Cressen and Evert [33] extensively described the vasculature in the stem of *E. viridis* Coville.

Khalaf et al. [5] studied the taxonomic characters of the *Ephedra* species in Egyptian flora based on morphometric and fatty acids characteristics and identified five *Ephedra* species (*E. alata* Decne., *E. aphylla* Forssk., *E. ciliata* Fisch. & C.A.Mey., *E. foeminea* Forssk, and *E. Ephedra pachyclada* Boiss.). In Egypt, each of these five species has male and female individuals. Among these five species, *E. alata* and *E. pachyclada* are categorized as “Least Concern Species”, while the other species are near threatened according to the IUCN [5]. The literature concerning the taxonomic description of male and female genders in the genus *Ephedra* is still scarce.

*Ephedra* species secure significant value across various domains including medicine, drug production, food, fodder, and the environment. It has been used for centuries in traditional medicine; ephedrine alkaloids were extracted particularly from *Ephedra sinaica* [34]. Ephedrine and pseudoephedrine are key ingredients in the pharmaceutical industry for the formulation of nasal congestion and respiratory disorder drugs [35]. *Ephedra* species are considered potential sources of natural flavors and livestock fodder [36]. From an environmental point of view, *Ephedra* species play a vital role in the arid and semi-arid ecosystem conservation as it contributes to soil stabilization and preventing erosion and is a potential candidate for phytoremediation [37].

However, these species possess only a few distinguishing morpho-taxonomical characteristics such as the presence of reproductive cones. Accordingly, taxonomists face problems in identifying such specimens that lack cones (stem fragments) for the sake of pharmacologists, forensic authorities, environmentalists, researchers, and others. Moreover, it is difficult for taxonomists to distinguish between species and genders in fragmented specimens or those without reproductive cones.

Our study is a unique approach that uses the stem anatomical and epidermal characteristics to distinguish between different *Ephedra* species at the gender level. This study aimed to (1) use the stem anatomical traits to identify the Egyptian *Ephedra* species, (2) distinguish between the genders in each species based on the stem anatomical and epidermal traits, and (3) deduce an anatomical key to distinguish genders and species.

## 2. Materials and Methods

### 2.1. Sampling and Study Area

A total of 120 fresh specimens representing the male and female individuals of the studied five *Ephedra* species (*E. alata* Decne., *E. aphylla* Forssk., *E. ciliata* Fisch. & C.A.Mey., *E. foeminea* Forssk, and *E. Ephedra pachyclada* Boiss.) were collected from South Sinai Wadis and mountains as shown in Table 1, and S. Sinai area was selected to unify the prevailing environmental conditions to study the anatomical features of the studied species. *Ephedra* species were identified by Professor Wafaa Amer in Cairo University Herbarium based on earlier floristic and taxonomic treatments [5,38,39,40,41], POWO, 2024 (https://powo.science.kew.org/taxon/urn:lsid:ipni.org:names:328160-2 accessed on 15 December 2023) and WFO, 2024 (https://www.worldfloraonline.org/taxon/wfo-4000013540 accessed on 19 March 2024). Synonyms were derived from the PLANT LIST (http://www.theplantlist.org/browse/G/Ephedraceae/Ephedra/ accessed on 15 January 2024), IPNI (https://www.ipni.org/ accessed on 8 June 2024), and WCVP (The World Checklist of Vascular Plants accessed on 20 July 2024).

### 2.2. Stem Anatomical Investigations

Sections (approximately 1–2 cm) from the third internode of the mature *Ephedra* stems were softened with a 50% glycerin solution (*v*/*v* aqueous solution) (ADWIC, Cairo, Egypt), then fixed in F.A.A. solution for 24 h to prevent tissue autolysis, and finally preserved in 70% ethanol (*v*/*v* aqueous solution) (ADWIC, Cairo, Egypt). Transverse cross sections (approximately 10–15 µm) were prepared using a REICHERT 820 Microtome (Tekyard LLC, Bloomington, CA, USA), double stained with a combination of safranine and light green (ADWIC, Cairo, Egypt), and then mounted on glass slides using oven-dried Canada balsam. The sections were examined and photographed using an OPTIKA light microscope (OPTIKA Srl, Ponteranica, Italy).

### 2.3. Epidermal Investigations

The anatomical characteristics were described following the anatomical description rules of Eames [42]. The epidermal stripes are mechanically prepared, stained with safranine and iodine solution, and mounted on slides using pure glycerin [43]. The examination was conducted using the 12.1-megapixel Canon PowerShot G15 (Canon Inc., Tokyo, Japan).

### 2.4. Statistical Analysis

The obtained data were analyzed using several software programs: R software version 4.3.2, Microsoft Excel 365, SPSS version 26.0, and Graph Pad Prism version 8.4.2. First, the data were checked for normality and homogeneity of variances using Kolmogorov–Smirnoff and Levene’s tests, respectively, in SPSS (IBM Corporation, Armonk, NY, USA). Since all data showed normality, one-way analysis of variance (ANOVA) followed by Tukey’s post hoc test analysis was used for all statistical comparisons. Graph Pad Prism software was then utilized to create a histogram plot. R software (Vienna, Austria) was used along with the “factoextra” and “FactoMineR” packages for principal component analysis (PCA) of a dataset containing continuous variables [44,45], based on 26 anatomical characters of the studied five *Ephedra* species (ten genders) as illustrated in Appendix A. The “ggplot2” package was employed to generate boxplots, illustrating the differences in the measured data [46]. Additionally, the “Corrplot” package was used to obtain and display the correlation coefficients for the interaction between two variables [47]. A strong positive correlation is depicted by a blue value of 1, while a white value of 0 indicates no relationship between the two variables. A significant negative correlation is represented by a pink value of −1. Finally, Microsoft Excel 365 was utilized to create radar and pie plots. (Statistical significance was defined as *p* < 0.05, and non-statistical significance as *p* > 0.05.)

## 3. Results and Discussion

### 3.1. Material Attributes

The genus *Ephedra* Tourn. ex L. possesses five species of perennial, dioecious under-shrubs, or shrubs in Egyptian flora. Khalaf et al. [5] studied the taxonomic characteristics of these medically economic species based on their morphometric and fatty acids characteristics. These species still exhibit taxonomic complications, especially for the specimens lacking the reproductive cones. In contrast, more complexity was attributed to the xeromorphic features of these species, where leaves are reduced in size and become membranous in mature plants, and the assimilating branches that replaced it functionally are shown in Figure 1. This study focused on the anatomical characteristics of the mature stem of five species to resolve its identification problem and extends beyond this to identify the genders of these species that are represented by ten individuals.

### 3.2. Anatomical Features of the Studied Stems

The anatomical characteristics play a crucial role in systematic and classification purposes, providing efficient evidence to identify the individual of uncertain taxonomic status [48,49,50].

Stem outline and cuticle: The studied stem sections appeared ridged and furrowed in male *Ephedra alata* and male *E. aphylla*, terete-ridged and furrowed in male *E. pachyclada*, and terete in the remaining seven studied individuals as illustrated in Figure 2. The foregoing data were congruent with Evans, Marsden, and Steeves, and Thoday and Berridge [19,51,52], who reported that the stem had a distinct wavy outline in some *Ephedra* spp. due to the variation in the presence of ridges and longitudinal furrows and that some *Ephedra* were terete. The epidermis of the studied ten individuals (two genders/species) appeared thick-walled and heavily cutinized. This finding was in harmony with Sharma et al., Marsden and Steeves, and Dörken [16,51,53].

Stem epidermis in T.S: The stem epidermis is a uniseriate or biseriate layer/s of thin-walled cells; it appeared uniseriate in male and female individuals of *E. alata*, *E. ciliata*, *E. foemina*, *E. pachyclada*, and female *E. aphylla*, while the male *E. aphylla* was distinguished by its biseriate epidermis as illustrated in Figure 2G,H. This finding matched the results reported by Sharma et al. [16]. Epidermal cell shape was radially elongated in *E. ciliata* (male and female) and in male *E. foemina*. Meanwhile, a radially and tangentially elongated epidermis was detected in female *E. foemina*; tangentially elongated epidermis was found in males and females of both *E. aphylla* and *E. pachyclada*, and tangentially and radially elongated epidermal cells were found in both genders of *E. alata*, as illustrated in Figure 2 and Appendix A.

Trichomes, papillae, and stomata: Non-glandular-unicellular, uniseriate trichomes with a swollen base (nearly cylindrical-shaped) were observed in female *E. aphylla* while non-glandular-unicellular, uniseriate trichomes with a wide base (flask-shaped) were observed in male and female *E. ciliata*, as shown in Figure 3A–F. These findings were in agreement with D. Pant and Verma [22]. While centrally or eccentrically located papillae were present in *E. alata*, *E. foemina*, *E. pachyclada* (male and female), and male *E. aphylla*. At the same time, they were not present in female *E. aphylla* and male and female *E. ciliata*, as shown in Figure 3. The given results agreed with Sharma et al., D. Pant and Verma, and Dörken [16,22,53] for the investigated *Ephedra* spp., including *E. alata*, *E. foemina*, and *E. pachyclada*. Sunken stomata were observed in all ten studied individuals as illustrated in Figure 2A–T and Appendix A, and this finding was in accordance with Evans, and Pant and Mehra [19,28].

Cortical tissue: The studied sectors showed that the number of cortical cell layers ranged from three to seven, and thin-walled chlorenchyma cells gave rise to two layers of elongated palisade cells and three layers of oval/circular spongy cells. Chlorenchyma is in 1–2 rows in female *E. aphylla*, *E. ciliata,* and *E. foemina*; 2–3 rows in male *E. aphylla* and *E. ciliata*; 3–4 rows in male *E. alata*, *E. foemina*, and *E. pachyclada* (male and female); or ill-defined number in female *E. alata* due to tanniniferous idioblast depositions. These results were found earlier by Sharma et al., Evans, and Thoday and Berridge [16,19,52]. Despite a low number of chlorenchyma rows in the studied photosynthetic stems its photosynthetic efficiency is high, as reported earlier by Ivanov et al. [54], who studied *E. sinaica* inhabiting Mongolian steppe, which has a low content of photosynthetic tissue with high photosynthetic efficiency and low transpiration rate, indicating remarkable water use capability.

Fiber patches were visualized in cortical tissue as illustrated in Figure 2 and Appendix A; these patches did not exceed 55 in female *E. aphylla* and male *E. alata*; reached 90 in *E. ciliata*; reached 100 in female *E. pachyclada*; exceeded that in male *E. aphylla*, *E. pachyclada,* and *E. foemina*; and were ill-defined in number in female *E. alata*. Thompson, Sharma et al., and Thoday and Berridge [8,16,52] reported that the fibers are found irregularly or in clusters of one to five rows in the *Ephedra* species cortex.

The cortical parenchyma was observed in 1–5 rows of polyhedral parenchyma in the studied ten individuals, as illustrated in Figure 2 and Appendix A. It was in 1-2 rows in male *E. alata* and females of *E. aphylla*, *E. foemina*, and *E. pachyclada*; 2–3 rows in male *E. ciliata*, *E. foemina*, and *E. pachyclada*; increased to 4–5 rows in male *E. aphylla* and were ill-defined in female *E. alata*. These foregoing data were in line with Thoday and Berridge [52]. Endodermis is well defined in male *E. alata*; male *E. aphylla*, *E. foemina*, and *E. pachyclada*; female *E. ciliata*; absent in female *E. aphylla*, *E. foemina*, and *E. pachyclada* and male *E. ciliata*; and ill-defined in female *E. alata*. This finding was consistent with Carlquist [32], who reported this in the Old-World *Ephedra* species.

The phloem fibers appeared as a continuous ring or as patches (discontinuous bundle caps) in *E. alata* (male and female), male *E. foemina*, and *E. pachyclada* (male and female) and continuous rings in the remaining five genders; this is similar to the results of Sharma et al. [16]. Behnke and Paliwal [55] observed only two cell types in the phloem of *E. foemina* C.A.Mey.

*Ephedra alata* pith is relatively wide, occupies 50% of the sector, and is composed of dead, thick-walled cells, many of which were filled with a hard brown substance. It is smaller in other *Ephedra* species and occupies 25% of the sector in *E. ciliata*. Pith area is wide in *E. alata* (male and female); male *E. aphylla*, *E. foemina*, *E. pachyclada* (male and female); and narrow in female *E. aphylla*, *E. foemina*, and *E. ciliata* (male and female). The pith cells were lignified and had normal parenchyma in *E. alata* (male and female), and female *E. aphylla* and *E. ciliata* had normal parenchyma (thin-walled) in the remaining six studied individuals. The foregoing data were in line with those obtained by Thompson [8]. Pith cavity was present in male *E. aphylla* and absent in the remaining nine studied genders as illustrated in Figure 2 and Appendix A.

Vascular system: Normal secondary growth was shown as a continuous siphonostelic structure in all ten studied individuals. Generally, tracheids in earlywood and latewood were difficult to distinguish. It is slightly wider in *E. aphylla* and *E. ciliata* (male and female); narrow in female *E. alata*, male *E. foemina*, and *E. pachyclada* (male and female); and moderate in the remaining two genders. Thompson, and Carlquist [8,56] reported compatible data. *Ephedra* possesses three kinds of tracheary elements: vessel elements, tracheids, and fiber-tracheids. The shorter tracheary elements are more adaptive at higher altitudes and to extreme drought conditions [57]. Additionally, xylem vessels were nearly absent as reported in *E. gerardiana* Wall. ex Stapf. and *E. rupestris* Benth compared to other *Ephedra* species [58,59,60].

The interfascicular regions are composed of secondary phloem, lignified conjunctive parenchyma, and some tracheids in female *E. alata*, male *E. foemina*, and *E. pachyclada* (male and female); they are composed of secondary xylem and secondary phloem in the remaining six studied genders. Tanniniferous idioblast cells are common in the cortex and pith in male and female *E. alata* and absent in the remaining eight studied genders [16]. Annual growth rings were noticed in male *E. alata* and *E. ciliata*, and absent in the remaining eight studied genders. Carlquist [32] reported that the growth ring in *Ephedra* species of the New World and multiple growth-ring types may develop inside a single stem in severe conditions. Stone cells of druses type in male and female individuals of *E. alata*, *E. aphylla*, *E. ciliata*, and *E. foemina* are illustrated in Figure 2 and Appendix A.

### 3.3. Stem Epidermal Characteristics

The shape of the epidermal cell is long and hexagonal in *E. aphylla* and *E. pachyclada* (male and female), female *E. ciliata*, and male *E. foemina*, and short and hexagonal in the remaining four studied individuals. Pant and Mehra [28] reported that the epidermal peels of *Ephedra* species stem have short, isodiametric, polygonal cells with straight anticlinal walls that are arranged unevenly. Pseudo-stripped stomata were observed in *E. alata* (male and female) and stripped in the remaining eight studied genders. The stomata shape was brachyparatetracytic in male individuals of *E. alata* and anomocytic in the remaining nine studied individuals. Pant and Mehra [28] reported similar relevant data. Calcium oxalate presents as solitary crystals distinguishing *E. alata* (male and female) and druses in the remaining eight studied genders. Carlquist [56] reported that calcium oxalate crystals were quite abundant in the wood of *E. pachyclada*, and *E. ciliata* as illustrated in Figure 4a–d and Appendix A. These studies were at the species level and there are no studies conducted on individuals at the “gender level”. Therefore, our study was considered unique as it was conducted at the gender level.

### 3.4. Key Distinguishing Species/Genders Based on Stem Anatomical and Epidermal Traits


**I-**
Tanniniferous idioblasts present
**
*E. alata*
**


**i-**
Terete outline, annual rings absent
**Female**


**ii**
Furrowed outline, annual rings present
**Male**


**+**
Tanniniferous idioblasts absent
**II**

**II-**
A large number of tracheids and wide, with or without trichomes
**III**

**III-**
Terete and furrowed outline, with or without trichomes, epidermis tangentially elongated
**
*E. aphylla*
**


**i-**
Terete outline, non-glandular-unicellular, uniseriate trichomes with a swollen base (nearly cylindrical-shaped) present
**Female**


**ii**
Ridged and furrowed outline, pith cavity present
**Male**


**+**
Terete outline, non-glandular-unicellular, uniseriate trichomes with a wide base (flask-shaped) present, epidermis radially elongated
**
*E. ciliata*
**

**i-**
Annual rings absent, presence of lignified and normal parenchyma in pith
**Female**


**ii-**
Annual rings present, only normal parenchyma in pith
**Male**

**IV**
Low number of tracheids and narrow, trichomes absent 
**V**

**V**
Number of protoxylem arches 10–15, terete outline, and epidermis radially elongated
**
*E. foemina*
**


**i-**
Ridges patches up to 1/2 of the cortex, xylem vessels occupy more than 50% of the sector 
**Female**


**ii-**
Ridges patches less than 20% of the cortex, xylem vessels occupy less than 25% of the sector 
**Male**


**+**
Number of protoxylem arches 10–12, terete or rigid and furrow outline, epidermis tangentially elongated
**
*E. pachyclada*
**

**i-**
Interfascicular cambium with restricted activity
**Female**

**ii-**
Interfascicular cambium with normal activity
**Male**


### 3.5. Anatomical Diversity at Interspecific and Intra-Generic Levels as Indicated by Principal Component Analysis

Figure 5 shows the PCA plot based on 26 anatomical characters of the studied five *Ephedra* species (two genders of each different species) as illustrated in Appendix A. At dimensions 1 and 2, with a cumulative variance of 48.8% for dimension 1, the clear separation of the studied genders into two groups along this dimension is observed. The right group includes five genders (female *E. alata* and *E. pachyclada* and male of *E. alata*, *E. pachyclada*, and *E. foemina*), while the left group represents the female *E. aphylla*, *E. ciliata*, and *E. foemina*, and male *E. aphylla* and *E. ciliata*. On the right side, the first group combined the male and female *E. alata* correlated with the presence of the tanniniferous idioblasts. The second group combined the male *E. pachyclada* with male *E. foemina* based on their larger number of chlorenchyma rows (3–4 rows) and the low number of tracheids (mean 4/patch). Whereas the female *E. pachyclada* was shared with the second group, this may be credited to having the same number of chlorenchyma rows (3–4 rows). On the left side, the first group combined the female *E. ciliata* with the male *E. aphylla* depending on the fact that they have the relevant numbers of rows of polyhedral parenchyma tissues (3–4, 4–5 rows, respectively) and a large number of protoxylem arches (15, 15–17, respectively). The radial and tangential elongation of the epidermis distinguishes female *E. foemina*. Notably, the presence of annual rings, trichomes, and a large number of protoxylem arches (15–17) distinguish male *E. ciliata*. Female *E. aphylla* was distinguished from the individuals of the left group due to the presence of the largest number of sunken stomata and tracheid arches/patches (75 and 9, respectively).

### 3.6. Diversity in the Mean Number of Tracheid Arches (Bundle) at Interspecific

The mean number of tracheid arches demonstrated significant interspecific variations. Figure 6 shows that the largest number of tracheid arches was recorded in female *E. aphylla*, followed by male *E. ciliata* (mean 9 and 8, respectively), while the lowest number was reported in the female *E. pachyclada* (mean 3). The dominance of tracheids in the studied plants may be supported by Hunziker and Riedl [58,61]. Moreover, the intra-gender variation was notable for each of the studied species. Nevertheless, the genders of each species were collected from the same population. However, this study did not focus on the tracheid pit types; the study carried out by Bauch et al. and Dute et al. [62,63] reported that *Ephedra* has pit membranes similar to those seen in conifer species.

### 3.7. Diversity in Stomata Density at Interspecific and Intra-Generic Levels as Shown by the Box Plot

Figure 7 demonstrates the mean number of sunken stomata per microscope field (300 µm) in the studied *Ephedra* species/genders to predict the species’ adaptation to drought in an arid environment. We found that female *E. aphylla* possesses the largest number of sunken stomata/microscope fields (mean 75/300 µm), followed by males of *E. aphylla* and *E. ciliata* (mean 60/300 µm). The lowest number was recorded in male *E. pachyclada* (mean 20/300 µm). In general, it was clear from the results that the male and female genders of *E. alata*, *E. aphylla*, and *E. ciliata* have a greater number of sunken stomata/microscope fields compared to the male and female genders of *E. foemina*, and *E. pachyclada*.

### 3.8. The Correlation Between the Anatomical Traits in the Studied Species/Genders

Figure 8 shows the Pearson′s correlation analysis carried out to explore the relationship among the anatomical characteristics in the studied *Ephedra* species/genders. The analyses showed significant positive correlations between some pairs, including annual rings–tanniniferous idioblasts; tracheid width–pith width; and tracheid arches–tracheid width. Significant negative correlations were also detected between other pairs, such as the presence of trichomes–the presence of papilla; sector outline number of sunken stomata–microscope field, and epidermis composition–pith cavity, as shown in Figure 8.

### 3.9. Diversity of Epidermal Characteristics at Interspecific and Intra-Generic Levels as Indicated by the Heat Map

Figure 9 presents the heat map of the epidermal characteristics of the studied species/genders of *Ephedra* visualized by color intensity into four clusters. The first cluster consisted of male individuals of *E. foemina* and *E. aphylla* depending on the fact that they have the largest length of stomatal and epidermal cells and have higher percentages of stomatal length and width. Female *E. pachyclada* is combined with this group because it also has a higher percentage of stomatal length and width. The second cluster grouped male *E. pachyclada* and female *E. aphylla* as they have the highest percentages in the stomatal index. The third cluster includes male *E. alata* and female *E. alata* based on their higher stomatal length and width, and also includes female *E. foemina* as it also has a higher stomatal length and width. The fourth cluster comprised male *E. ciliata* and female *E. ciliata* based on the absence of papilla and having higher values of epidermal cell width.

### 3.10. Diversity of the Epidermal Cell Size at Interspecific and Intra-Generic Levels as Indicated by the Radar Plot

This study outlined that the epidermal cell size (L × W µm), as shown in Figure 10, played an important role in *Ephedra* identification. Male *E. foemina* possesses the highest cell length (10.9 µm) followed by male *E. aphylla* (9.3 µm), while the lowest length was recorded in female *E. alata* (4.4 µm). On the other hand, the width of the epidermal cell was the largest in male *E. alata* (5.17 µm) followed by female *E. foemina* (3.48 µm), while the lowest width was recorded in male *E. foemina* (1.71 µm). Based on the previous results, there is a clear inverse relationship between the length and width of the epidermal cells.

### 3.11. Diversity in Stomatal Index at Interspecific and Intra-Generic Levels

The stomatal index was calculated as a percentage and outlined in Figure 11. The highest stomatal index percentages were recorded in male *E. pachyclada* (18.05%) followed by females of *E. aphylla* and *E. ciliata* (15.12, and 14.13%, respectively), and these data were in accordance with D. Pant and Verma [22]. Meanwhile, the lowest percentage was reported in female *E. foemina* (2.17%). Furthermore, there was substantial intra-species variance for every species; nonetheless, all species’ genders were gathered from the same population.

### 3.12. The Correlation Between the Epidermal Traits in the Studied Species/Genders

Pearson’s correlation analysis, as shown in Figure 12, was carried out to explore the relationship between the used epidermal characteristics in the studied *Ephedra* species/genders. The analyses showed significant positive correlations between some pairs: epidermal cell shape–stomatal index; epidermal cell length–stomatal length; crystals type–crystals abundance; and stomatal type-stomatal width. Significant negative correlations were also observed between epidermal cell shape—stomatal width; epidermal cell length–epidermal cell width; stomatal width—stomatal lengths; and stomatal width—stomatal index, as shown in Figure 12. The achieved results support the importance of epidermal features as a tool to distinguish between the studied *Ephedra* species at the gender level.

## 4. Conclusions

The taxonomists face identification challenges when *Ephedra* specimens lack characteristic reproductive cones. These challenges can be overcome using stem anatomy. This current study has revealed that stem anatomy was an efficient tool for species identification of the *Ephedra* specimens up to the gender level, including distinctive characteristics such as the tanniniferous idioblasts, stem outline, number of tracheids, and number of protoxylem arches. Multidisciplinary approaches should be applied simultaneously to identify the specimens with reduced morphological features. Our findings have significant implications for researchers, pharmacists, and environmentalists who need accurate identifications of *Ephedra* specimens for their work. Furthermore, this study encourages future investigations into the anatomical features of other *Ephedra* species, potentially leading to a broader understanding of the genus and its taxonomic complexities.

## Figures and Tables

**Figure 1 biology-13-00947-f001:**
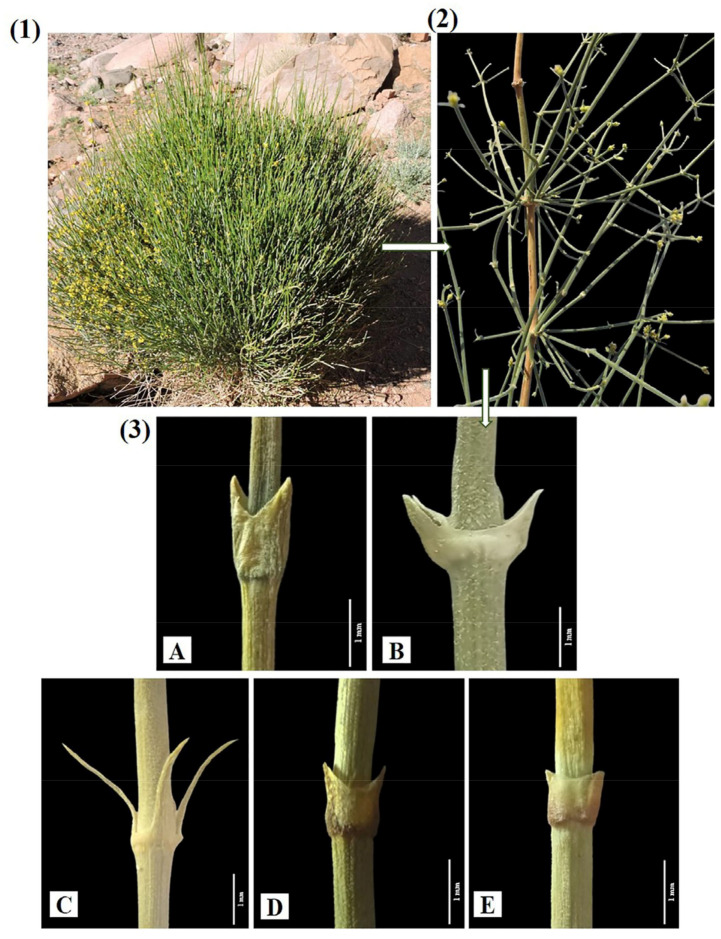
(**1**) Xeromorphic habit of the *Ephedra* in the field. (**2**) Image of an enlarged part of the stems (**3**) Images of assimilating branches showing reduced leaves of the studied species. (**A**) *E. alata*; (**B**) *E. aphylla*; (**C**) *E. ciliata*; (**D**) *E. foemina*; and (**E**) *E. pachyclada*.

**Figure 2 biology-13-00947-f002:**
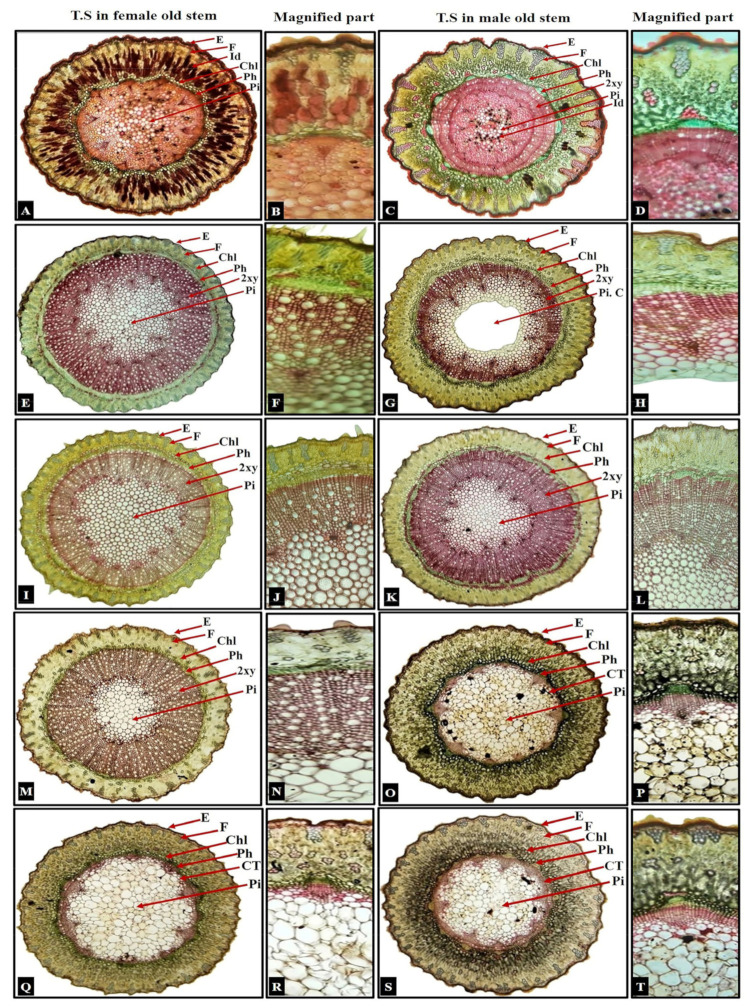
T.S. in the studied ten *Ephedra* stems demonstrating the differential anatomical features. (**A**–**D**) *E. alata*; (**E**–**H**) *E. aphylla*; (**I**–**L**) *E. ciliata*; (**M**–**P**) *E. foemina*; and (**Q**–**T**) *E. pachyclada*. E = epidermis; F = fiber patches; Chl = chlorenchyma; Ph = phloem; 2xy = secondary xylem; Pi = pith; Pi. C = pith cavity; Id = idioblasts; CT = conjunctive tissue. (T. S in the stem at 4×; magnified part at 10×.).

**Figure 3 biology-13-00947-f003:**
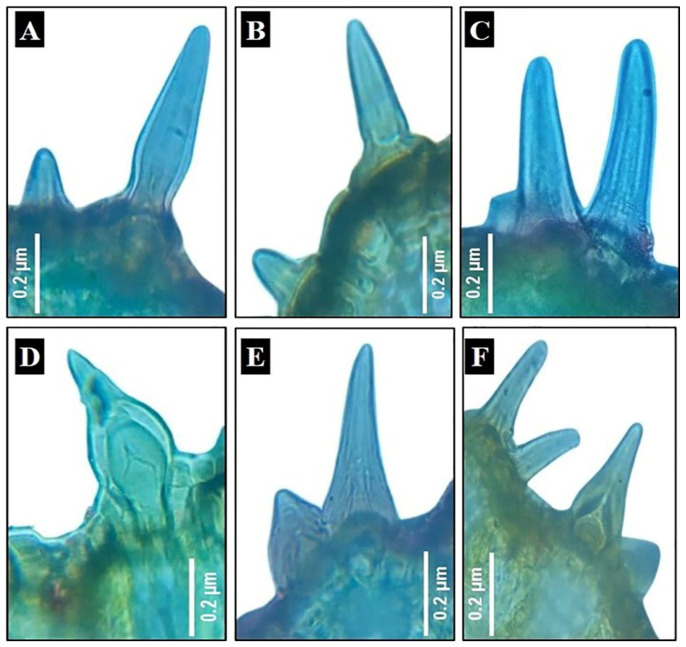
Trichomes of *Ephedra* stem of the studied species/genders. (**A**,**B**) *E. aphylla* female; (**C**,**D**) *E. ciliata* female; and (**E**,**F**) *E. ciliata* male.

**Figure 4 biology-13-00947-f004:**
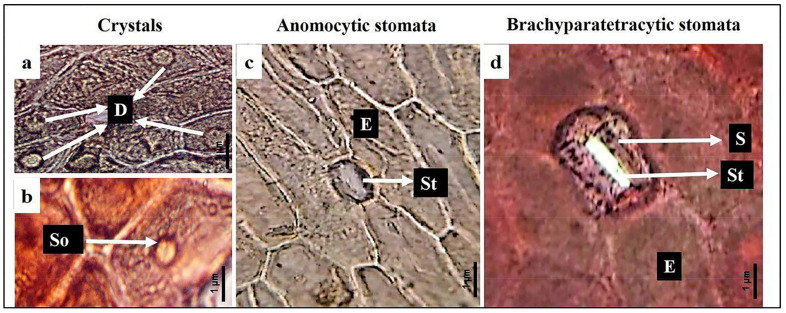
Features of the epidermal system in the studied *Ephedra* species. (**a**) Druses crystal in male *E. ciliata*; (**b**) solitary crystal in female *E. alata*; (**c**) anomocytic stomata in male *E. aphylla*; (**d**) brachyparacytic in male *E. alata*. D = druses; So = solitary; E = epidermal cell; St = stomata; and S = subsidiary cell.

**Figure 5 biology-13-00947-f005:**
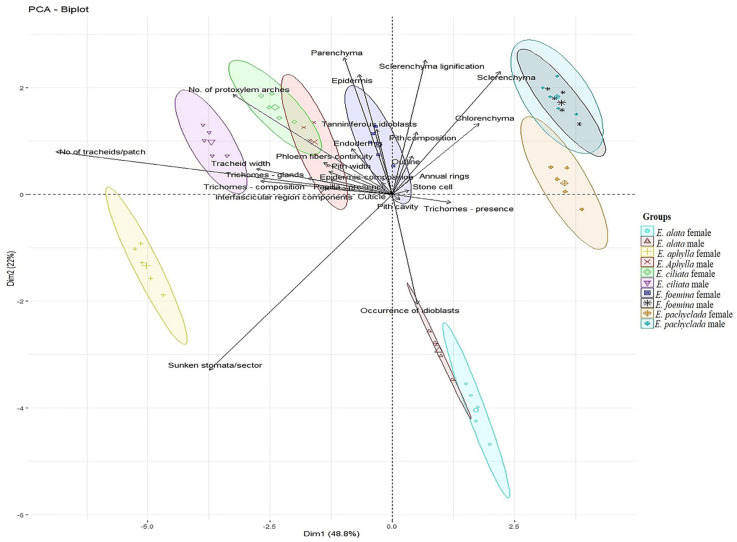
Principal component analysis (PCA) of the 26 anatomical characteristics with 48.8% total variance described along the first axis (Dim 1) followed by 22% variance exhibited along the second axis (Dim 2).

**Figure 6 biology-13-00947-f006:**
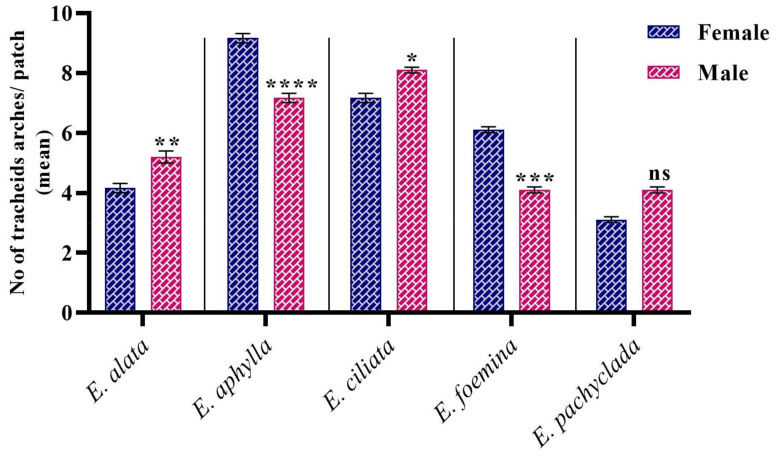
The mean number of tracheid arches (bundles) in the studied *Ephedra* species/genders. *p* value: * *p* ≤ 0.05, ** *p* ≤ 0.01, *** *p* ≤ 0.001, **** *p* ≤ 0.0001. ns: not significant.

**Figure 7 biology-13-00947-f007:**
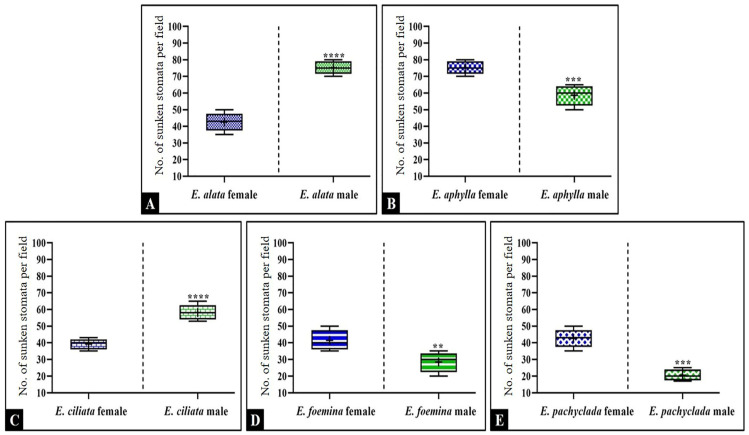
The box plot demonstrates the mean number of sunken stomata/microscope fields (300 µm) in the studied *Ephedra* species/genders. *p* value: ** *p* ≤ 0.01, *** *p* ≤ 0.001, and **** *p* ≤ 0.0001. (**A**) *E. alata*; (**B**) *E. aphylla*; (**C**) *E. ciliata*; (**D**) *E. foemina*; and (**E**) *E. pachyclada*.

**Figure 8 biology-13-00947-f008:**
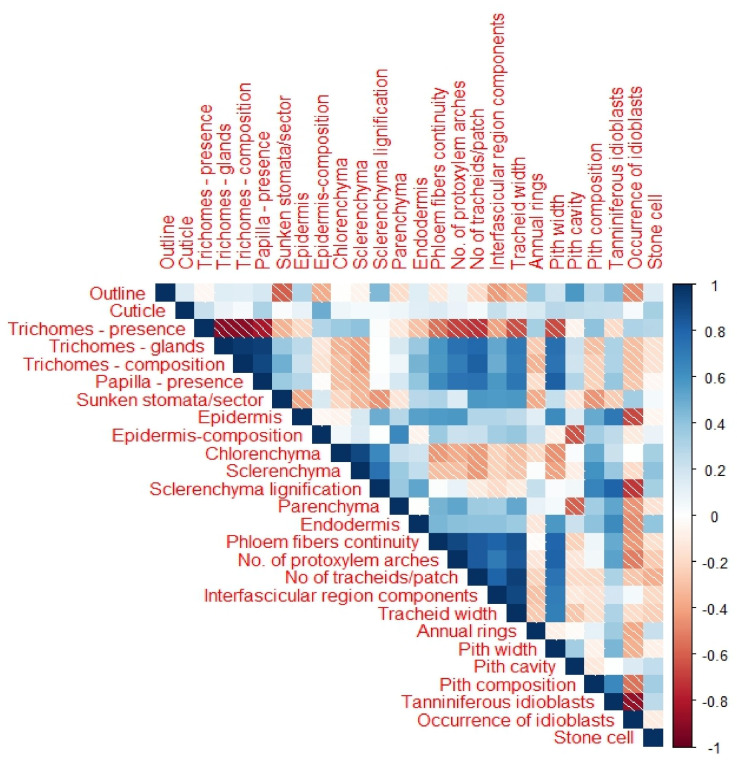
Pearson′s correlation analysis is based on the correlation coefficient of anatomical characteristics of the studied *Ephedra* species. Positive and negative correlations are displayed in blue and red colors, respectively. Correlation coefficients are proportional to color intensity. (///) indicates negative correlations.

**Figure 9 biology-13-00947-f009:**
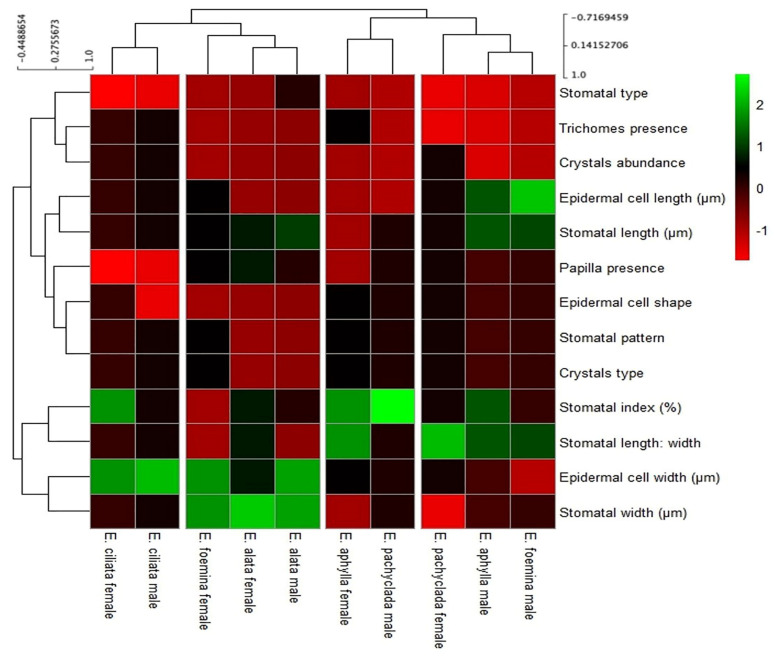
Hierarchical clustering analysis of epidermal characteristics of the studied species/genders of *Ephedra*. Data are represented by the means of 3 replicates.

**Figure 10 biology-13-00947-f010:**
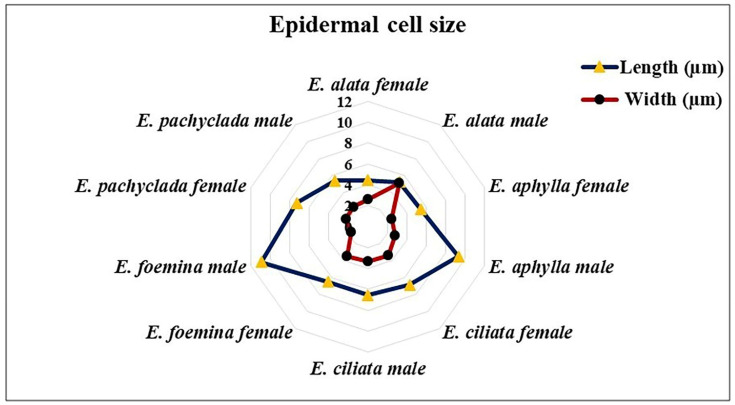
Radar plot of the epidermal cell size (L × W µm) in the studied *Ephedra* species at the gender level.

**Figure 11 biology-13-00947-f011:**
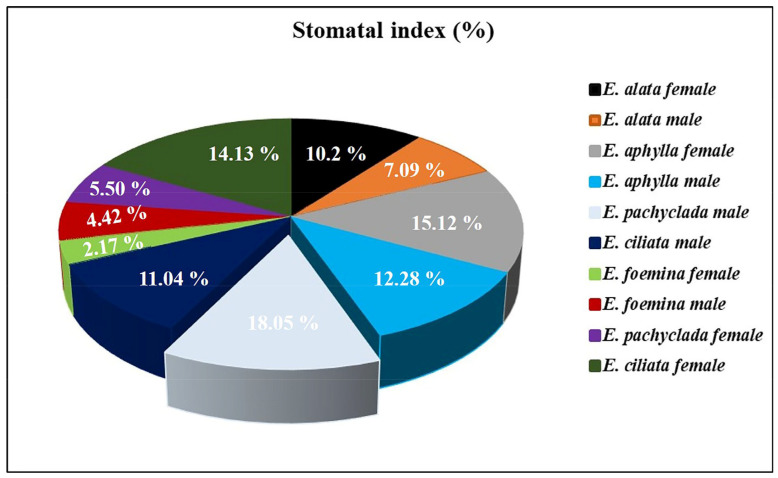
Pie chart of the stomatal index (%) in the studied *Ephedra* species at the gender level.

**Figure 12 biology-13-00947-f012:**
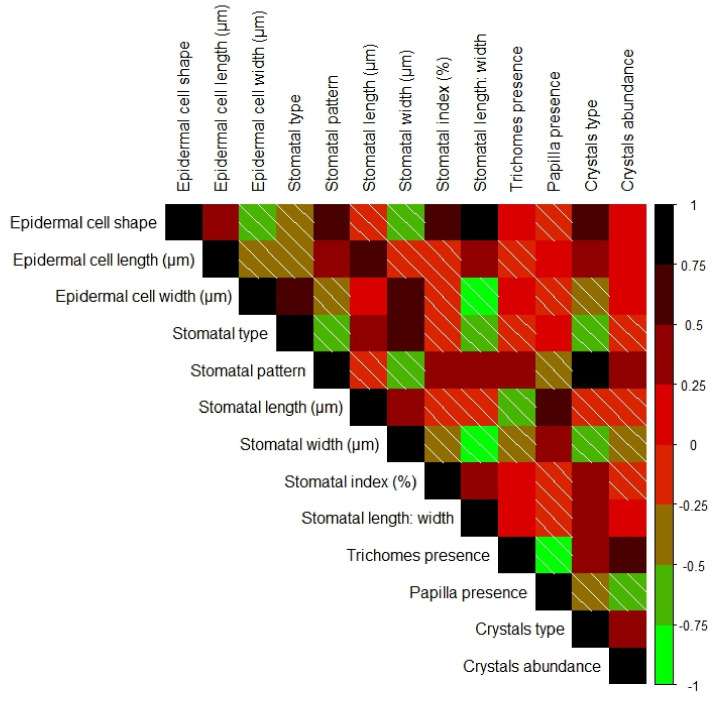
Pearson’s correlation analysis is based on the correlation coefficient of epidermal characteristics of the studied *Ephedra* species. Positive and negative correlations are displayed in black and green colors, respectively. Correlation coefficients are proportional to color intensity. (///) means negative correlations.

**Table 1 biology-13-00947-t001:** Locations of the study *Ephedra* samples in S. Sinai.

Species	Locality	GPS Coordinates		Gender	Date of Collection
		N	E		
*E. alata*	Abo Zeinema, Sinai	29°02′31″	33°06′30″	Male	14 June 2023
	Wadi Feiran, Sinai	28°43′06″	33°37′06″	Female	15 May 2024
*E. aphylla*	Wadi Al-Arbaeen Saint Catherine, Sinai	28°55′88″	33°94′98″	Female and Male	17 August 2023
*E. ciliata*	Wadi Lethi, Saint Catherine, Sinai	28°33′07″	33°58′24″	Male	15 July 2023
	The garden of. St, Catherine’s Monastery, Sinai	28°33′25″	33°58′23″	Female	25 August 2023
*E. foemina*	Gabal Mousa, Sinai	28°54′13″	33°97′55″	Female and Male	16 July 2023
*E. pachyclada*	Abu Walia, Saint Catherine, Sinai at 1905 altitude	28°53′55″	33°91′39″	Female	16 August 2023
	Abu Walia, Saint Catherine, Sinai at 1891 altitude	28°53′60″	33°90′92″	Male	16 August 2023

## Data Availability

Data are contained within the article.

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
