# Peer review of "Anatomical Tool as Additional Approach for Identifying Pharmaceutically Important Ephedra Species (Ephedraceae) at Gender Identity Level in Egypt"

_biology, 2024, doi:10.3390/biology13110947_

Round 1
Reviewer 1 Report
Comments and Suggestions for Authors
Dear authors, the manuscript “Anatomical tool as an additional approach for identifying the pharmaceutically important Ephedra species at the gender level in Egypt” needs further revisions*.
*All considerations, suggestions and/or corrections are highlighted in the attached PDF file.

Author Response
Replies to Reviewer 1
Dear appreciated reviewer 1
Thank you for your thoughtful comments and suggestions for improving our manuscript. We agree with all of your points and made the changes you suggested to the manuscript.
Note: The corrections of reviewer 1 were highlighted in green color in the manuscript.
Comments and Suggestions for Authors:
Point 1: In the title, Line 2, it is important to indicate in parentheses the name of the taxonomic family to which Ephedra belongs in the manuscript title.
Response 1: Done. Anatomical tool as an additional approach for identifying the pharmaceutically important Ephedra species (Ephedraceae) at gender identity level in Egypt.
Point 2: In the keywords, Line 46, avoid using words that are present in the manuscript title.
Response 2: Done. Ephedra fragments; light microscope; stem anatomy; epidermal system; gender identification; Ephedraceae.
Point 3: In the introduction section, Line 57, avoid repeating words, go straight to the name of the genus.
Response 3: Done. Deleted. The Ephedra includes small, drought-resistant.
Point 4: In the results and discussion section, lines 186-188, indicate to the reader which images in Figure 2 represent this description.
Response 4: Done. it appeared uniseriate in male and female individuals of E. alata, E. ciliata, E. foemina, E. pachyclada, and female E. aphylla, while, the male E. aphylla was distinguished by its biseriate epidermis as illustrated in Figure 2H,G.
Point 5: In the results and discussion section, line 193, it is important to identify in the image captions which are female and which are male. Make this clear to the reader.
Response 5: It is outlined at the top of the image (Figure 2)
Point 6: In the results and discussion section, line 195, there is no caption for the species in Figure 3. It is not possible to know which species these characteristics belong to.
Response 6: Done. The caption has been modified to “Figure 3. Trichomes of Ephedra stem of the studied species/genders. (A, B): E. aphylla female; (C, D): E. ciliata female; and (E, F): E. ciliata male”.
Point 7: In the results and discussion section, line 202, indicate to the reader which images in Figure 2 represent this description.
Response 7: Done. Sunken stomata were observed in all ten studied individuals as illustrated in Figure 2A-T.
Point 8: In the results and discussion section, figure 3, line 286, add the caption that identifies the species related to these images.
Response 8: Done. The caption has been modified to “Figure 3. Trichomes of Ephedra stem of the studied species/genders. (A, B): E. aphylla female; (C, D): E. ciliata female; and (E, F): E. ciliata male”.
Point 9: In the results and discussion section, line 300, see previous comment. About the citation of images in the description.
Response 9: Done. The caption has been modified to Carlquist [56] reported that calcium oxalate crystals were quite abundant in the wood of E. pachyclada, and E. ciliata as illustrated in Figure 4a-d and Supplementary Table S3.
Point 10: In the results and discussion section, line 337, what anatomical characters?
Make it clear to the reader in the M&M.
Response 10: Done. Figure 5 showed the PCA plot based on 26 anatomical characters of the studied five Ephedra species (ten genders) as illustrated in Supplementary Table S2, and it was modified in the materials and methods section to r software (Vienna, Austria) was used along with the "factoextra" and "FactoMineR" packages for principal component analysis (PCA) of a dataset containing continuous var-iables [44,45], based on 26 anatomical characters of the studied five Ephedra species (ten genders) as illustrated in Supplementary Table S2 lines 144-148.
Point 11: In the results and discussion section, line 381, Ephedra is in italic here.
Response 11: Done. Ephedra.
Point 12: In the conclusions section, Ephedra is in italic in lines 535, 538, 543, and 544.
Response 12: Done.
Please accept my respect
Maha H Khalaf

Reviewer 2 Report
Comments and Suggestions for Authors
The morphological approach is traditional approach and should be incorporated with advance technique to attract reader. The summary was mislead due to no fatty acid finding in the manuscripts. Several grammatical error detected. Include chromosomal study to strengthen the manuscript finding.

Author Response
Replies to Reviewer 2
Dear appreciated reviewer 2
Thank you for your thoughtful comments and suggestions for improving our manuscript. We agree with all of your points and made the changes you suggested in the manuscript.
Note: The corrections of reviewer 2 were highlighted in yellow color in the manuscript.
- Open Review
Point 1: The methods must be improved.
Response 1: Done. The methods were improved.
Point 2: The conclusions must be improved.
Response 2: Done. The conclusions were improved.
- Comments and Suggestions for Authors
Point 3: The morphological approach is traditional approach and should be incorporated with advance technique to attract reader.
Response 3: You are right; it is a fact that the morphological and anatomical approaches are traditional and easily applied tools. Accordingly, these tools were selected to identify the ephedra species to the gender level, to adjust and cope with the nature of the commercial material (Ephedra fragments) used for Ephedrine production, and other pharmaceutical preparations, to help the scientists, traders, and the quality control in the pharmaceutical sector
Point 4: The summary was mislead due to no fatty acid finding in the manuscripts.
Response 4: The word fatty acids were removed and the summary reformed, but it was previously studied by the authors in this manuscript. https://doi.org/10.3390/plants13172442
Point 5: Several grammatical errors detected.
Response 5: The manuscript was reviewed and several grammar mistakes were corrected.
Point 6: Include chromosomal study to strengthen the manuscript finding.
Response 6: This is our plan for the rest of the thesis study. But this needs fresh rootlets or viable seeds, and this is not available in our harsh desert environment we still missing the adequate rainfall from years ago to improve the vegetative growth of these species.
Point 7: In the title, Line 3, needs to specify these or change to suitable words.
Response 7: Done. Anatomical tool as an additional approach for identifying the pharmaceutically important Ephedra species (Ephedraceae) at gender identity level in Egypt.
Point 8: In the simple summary, Line 14, explain the importance of these plants in medicine and environmental.
Response 8: It is outlined in lines (86-94) in the introduction section.
Point 9: In the simple summary, Line 15-16, traditional identification approach pose a lot of challenging; thus focus only for plant morphology not justifiable.
Response 9: Reformed. Based on the morphological characteristics, taxonomists often face challenges in identifying Ephedra specimens lacking reproductive cones, which are key features for identification.
Point 10: In the simple summary, Lines 19-20, how this finding could help this targeted expertise?
Response 10: Done, Modified to our findings revealed significant differences in these features among these are the tanniniferous idioblasts, stem outline, number of tracheids, and number of protoxylem arches, allowing for reliable identification of each species and genders within each species.
Point 11: In the simple summary, Lines 23-24, no fatty acid analysis is presented here.
Response 11: I am sorry the deletion and modification have been done. Our findings suggest that a multidisciplinary approach, combining anatomy with morphological characteristics, can effectively identify Ephedra specimens with reduced morphological features.
Point 12: In the abstract section, Line 44, how this possible if the fragment is dry?
Response 12: We used the dry fragments and herbarium materials for anatomical study by softened with a 50% glycerin solution (v/v aqueous solution), then fixed in F.A.A. solution for 24 hours to prevent tissue autolysis, and finally preserved in 70% ethanol (v/v aqueous solution). As mentioned in the materials and methods section lines 125-127.
Point 13: In the introduction section, Line 84, replace “is” with “was”.
Response 13: Done. The literature was still very poor.
Point 14: In the introduction section, Line 84, suggest to replace this word with the more convincing sentences.
Response 14: Done. It was replaced by the literature was still very poor concerning the taxonomic description of male and female genders in the genus Ephedra.
Point 15: In the materials and methods section, Line 113, state the name of the person that identify the species and number of voucher.
Response 15: Ephedra species were identified by Professor Wafaa Amer in Cairo University Herbarium, and the voucher specimens deposited in Cairo University Herbarium were arranged without numbers according to Engler system.
Point 16: In the materials and methods section, Lines 125-126, add (v/v) unit.
Response 16: Done. With a 50% glycerin solution (v/v aqueous solution), then fixed in F.A.A. solution for 24 hours to prevent tissue autolysis, and finally preserved in 70% ethanol (v/v aqueous solution).
Point 17: In the materials and methods section, Line 129, add full stops and rephrase the sentence.
Response 17: Done. The sections were examined and photographed by using an OPTIKA light microscope.
Point 18: In figure 1, the arrow should be smaller.
Response 18: Done in the figure.
Point 19: In the results and discussion section, line 186, the arrangement is not clear from the figure.
Response 19: The uniseriate or biseriate layers in (figure 2 H, J) appeared clear under the light microscope with high magnifications.
Point 20: In the results and discussion section, lines 232-234, gave the size.
Response 20: Done. Ephedra alata pith is relatively wide, occupies 50% of the sector, and is composed of dead, thick-walled cells, many of which were filled with a hard brown substance, and smaller in other Ephedra species, occupies 25% of the sector as in E. ciliata.
Point 21: In the results and discussion section, line 291, replace “is” with “was”.
Response 21: Done. Stomata shape was Brachy-paratetracytic.
Point 22: In the results and discussion section, line 298, replace “is” with “was”.
Response 22: Done. Therefore our study was considered.
Point 23: In the results and discussion section, line 341, replace “is” with “was”.
Response 23: Done. Whereas the female E. pachyclada was shared.
Point 24: In the results and discussion section, line 348, replace “is” with “was”.
Response 24: Done. E. aphylla was distinguished.
Point 25: In figure 5, italic for scientific name.
Response 25: Done in the figure.
Point 26: In the results and discussion section, line 369, replace “shows” with “showed”.
Response 26: Done. Figure 6 showed that.
Point 27: In the results and discussion section, line 425, replace “are” with “were”.
Response 27: Done. Among them were the annual rings.
Point 28: In figure 8, the font are too big for figure.
Response 28: Done in the figure.
Point 29: In the conclusions section, suggest to write this paragraph and summarize the key pointbfrom the finding above.
Response 29: Done.
Please accept my respect
Maha H. Khalaf

Reviewer 3 Report
Comments and Suggestions for Authors
The research is targeted to find anatomical differences between Ephedra species to ease the taxonomic and gender differentiation of this genus members. Besides academic importance, it is important for pharmaceutical and ecological purposes. The paper clearly states the goals of the research, adequately describes the methods and the results. The authors demonstrate the knowledge of literature in introduction and throughout the manuscript. The figures are informative and of high quality. Comments: Please expand on figure 3 legend: write the species and genders shown in figure 3A-F. Please proofread for minor typos in the text.
Author Response
Dear appreciated reviewer 3
Thank you for your thoughtful comments and suggestions for improving our manuscript. We agree with all of your points and made the changes you suggested to the manuscript. Note: The corrections of reviewer 3 were highlighted in turquoise color in the manuscript.
Point 1: Please expand on figure 3 legend: write the species and genders shown in figure 3A-F. Response 1: Done. The caption has been modified to “Figure 3. Trichomes of Ephedra stem of the studied species/genders. (A, B): E. aphylla female; (C, D): E. ciliata female; and (E, F): E. ciliata male”.
Point 2: Please proofread for minor typos in the text.Response 2: Done. The manuscript was reviewed and several grammar mistakes were corrected.
Point 3: Lines 95-100 clearly states that without reproductive cones there are no anatomical means to distinguish Ephedra species. Other means to distinguish the Ephedra species by anatomical traits is required. Although anatomical description was carried out in the past (publications 30-33), the differentiation by gender was not performed.
Response 3: Lines 95-100 are part of the introduction. In this part, we indicate that before our study, researchers could not distinguish between Ephedra species that lake reproductive cones using morphological methods. I indicated in this revised version that I mean morpho-taxonomical characteristics. So, in our study, we aim to distinguish the Ephedra species by anatomical traits to help pharmacologists, traders, forensic researchers, and others who used the crushed, or fragmented materials. Anatomical description was carried out in past publications but based on their results they couldn’t differentiate by gender. So, our study aims to find more anatomical traits that can be differentiated by gender.
Point 4: The microscopic methods for the anatomical Ephedra genus differentiation require specialized equipment and skills. It is not suitable for the fieldwork. The genetic methods might be a better and more precise option (for example, the methods described in DOI: 10.5586/asbp.3497). However, refining the knowledge of Ephedra species anatomy is still adding to the knowledge in the field. It can be useful for researchers who are more familiar and have access to microscopes and microtomes, but not to molecular biology equipment.
Response 4: The equipment required for microscopic methods for anatomical differentiation is available at most taxonomic laboratories. Microscopic methods could be done easily at laboratories and take a short time. Also, many researchers and technicians have the required skills. For researchers, the anatomical skills can be acquired easily. However, genetic methods are among the appropriate and recent methods for identification. However, fresh material or viable seeds are needed, which are not available in our harsh desert (lacking adequate rainfall) environment. So, the anatomical approach is a well-documented method and requires less equipment and skills than the genetic approach.
Point 5: Line 108 - is total of 20 plants sufficient number for drawing conclusions for 5 species, 2 genders (2 plants per species and gender)? Please clearly describe the number of analyzed sections and fields of view for all applicable figures.
Response 5: Thanks for this valuable question. It is indeed a mistake from the authors when writing the number, it is 120 individuals (24 for each species; each gender represented by 12 individuals), and it was already modified within the manuscript.
Point 6: Figure 3 - please add information about trichomes for all species. The difference between the species is not clear from the images. However, it is important for the Key 3.4.
Response 6: Thanks for this valuable comment. More description was added based on your recommendation. Added in lines 195-198 and added in the Key 3.4 in lines 322, and 325.
Point 7: conclusions are consistent with the data. However, the authors state general conclusions in the conclusions and abstract sections. It would be more useful for a reader to see the brief description of the differences in anatomical traits of studied species (some kind of synopsis of 3.4 section).
Response 7: Thanks for this valuable comment. Modified. A brief description of the differences in anatomical traits of studied species was added to the abstract lines 34-37 and conclusions lines 537-541.
Point 8: Line 83-84 - elucidate the status of other species.
Response 8: Thanks for this valuable comment. Done. I indicated in the revised version the status of other species as near threatened.
Point 9: line 173-175, line 182-185 -add reference to a figure demonstrating it.
Response 9: Thanks for this valuable comment. Done in line 179 and line 189.
Point 10: Figure 1 looks stretched along the horizontal axis (the letters are misshaped) - please correct it.
Response 10: Thanks for this valuable comment. Done.
Point 11: Figure 2, 4 - please add a scale bar.
Response 11: Thanks for this valuable comment. Done. The magnification power has been added under the legend of figure 2 and the scale bar has been added to figure 4.
Point 12: Line 333, 335 - two genders of different species, not ten or five genders.
Response 12: Thanks for this valuable comment. Done.
Point 13: Line 521-change overwhelmed to overcome.
Response 13: Thanks for this valuable comment. Done.
Point 14: Reference 3 - correct the authors' names (change the first letters to capital).
Response 14: Thanks for this valuable comment. Done.
Please accept my respect
Maha H Khalaf
